# Environmental Degradation of Plastic Composites with Natural Fillers—A Review

**DOI:** 10.3390/polym12010166

**Published:** 2020-01-08

**Authors:** Mihai Brebu

**Affiliations:** “Petru Poni” Institute of Macromolecular Chemistry, 41A Grigore Ghica Voda Alley, 700487 Iasi, Romania; bmihai@icmpp.ro

**Keywords:** polymer matrix, natural fibers, biodegradation, reprocessing, ageing, weathering

## Abstract

Polymer composites are widely used modern-day materials, specially designed to combine good mechanical properties and low density, resulting in a high tensile strength-to-weight ratio. However, materials for outdoor use suffer from the negative effects of environmental factors, loosing properties in various degrees. In particular, natural fillers (particulates or fibers) or components induce biodegradability in the otherwise bio inert matrix of usual commodity plastics. Here we present some aspects found in recent literature related to the effect of aggressive factors such as temperature, mechanical forces, solar radiation, humidity, and biological attack on the properties of plastic composites containing natural fillers.

## 1. Introduction

Environmental degradation of polymeric materials can occur by physical, chemical, and biological processes or a combination of them, under the effect of several factors such as temperature (thermal degradation), air (oxidative degradation), moisture (hydrolytic degradation), microorganisms (biodegradation), light (photo degradation), high-energy radiation (UV, γ irradiation), chemical agents (corrosion), mechanical stress [1]. These factors lead to irreversible changes in the material, which usually occurs in two phases. Deterioration of appearance (e.g., color change), physical, and morphological properties (e.g., mass change, crystallinity) or mechanical properties (e.g., embrittlement, fragmentation) occurs first (disintegration), and eventually the material completely breaks down into water, carbon dioxide, and other simple inorganic compounds (mineralization). All materials are more or less affected by the above-mentioned factors; however, it is how fast the processes occur that distinguish the environmentally degradable materials from the non-degradable ones. In particular, the evolution of the process in the initial stage (lag phase) is the time determining step for degradation. Environmental degradation is a desired process in the case of waste removal, but its effects are unwanted when they induce loss of properties in materials for outdoor use.

Here we present an overview of the latest reports on the effect of the most common environmental factors on the degradation of plastic composites with natural fillers. We first introduce some general aspects of environmental degradation of materials, defining the terms and testing practices. Then we present some generalities on polymer matrices and on natural fillers used in composites. The alteration of mechanical properties under the effect of repeated processing is described, with particular examples. Biodegradability is discussed, with examples of composites with bio-inert or with biodegradable polymeric matrices, emphasizing the role of matrix/filler interface and describing different degradation stages. The effect of photoradiation and of moisture is discussed, with particular examples, in the section dedicated to weathering and accelerated ageing of polymer composites with natural fillers.

## 2. Generalities of Environmental Degradation of Materials

Two approaches are usually considered in the design of environmentally degradable polymeric materials in an attempt to reduce the global problem related to inert polymer waste. One is to design polymeric materials with inherent biodegradability, and another is to enhance the biodegradation of recalcitrant petroleum-based polymers by modifying them (e.g., in blends or composites) with degradable components, usually bio-based ones, that can induce degradation by generating free radicals [2]. Examples of biodegradable polymeric materials include starch, chitosan, chitin, cellulose, lignin, polylactic acid (PLA), poly(3-hydroxybutyrate) (PHB), poly(3-hydroxybutirate-3-hydroxyvalerate) (PHBV), poly(butyrate adipate-co-terephthalate) (PBAT), poly-ε-caprolactone. A clear distinction should be made between bio-based polymers and biodegradable ones, since some biodegradable plastics are made from fossil resources, while some plastics made from biomass are non-biodegradable. It is also important to consider the specific conditions and the timeframe under which a ‘biodegradable’ polymer actually biodegrades. For example, most packaging materials marked as ‘biodegradable’ completely break down only if composted in industrial units, while they will most probably have limited biodegradation when landfilling. It is more difficult to establish the reasonable period of time in which the changes in the material allows it to be considered biodegradable, since this varies widely depending on the product, application, and environmental conditions. For example, it is desirable for plastic mulch film to completely degrade before the following crop cycle, to avoid soil burial of incompletely degraded plastic fragments [3].

Environmental degradation and the toxicity level of plastic materials is determined based on various standard methods and testing practices [4], as briefly described below.

Ultimate biodegradation follows the evolution of CO_2_ and CH_4_ when polymeric materials are maintained in microbial conditions. Respirometric test methods were standardized both for aerobic biodegradation in soil burial [5,6] or in compost [7], and for anaerobic biodegradation under sewage sludge [8] or anaerobic digestion [9]. While these methods are similar in the procedure of measuring the evolving CO_2_ and CH_4_, they differ in testing conditions, substrate composition, and type of microbial inoculums used for tests.

Ecotoxicity determines the potential environmental toxicity of all products (e.g., volatile gases, leachate, residue) resulting from the biodegradation or composting processes. The large macromolecular backbone of polymers is virtually harmless, being not directly available to living cells; however, low-molecular mass compounds such as additives, degradation products, and intermediates (e.g., oligomers, monomers) or metabolic derivatives, can be harmful to the living organisms in the environment [10,11].

Changes in properties of the material such as changes of the molecular mass and its distribution (determined by size exclusion chromatography—SEC); structural and compositional changes of chemical species in the material or its degradation products (determined by infrared spectroscopy—IR or nuclear magnetic resonance spectroscopy—NMR); changes in physical and morphology properties, such as surface features, mass loss, glass transition (Tg), melting temperature (Tm), crystallinity, thermal behavior (determined by differential scanning calorimetry—DSC or thermogravimetry—TG thermal analysis methods); or modification in mechanical properties, such as tensile strength or elongation at break, are useful parameters to evaluate the degradability of materials. The above-mentioned analysis and characterization methods are suitable to evaluate the evolution of degradation; however, they cannot be used for direct quantification of the processes. For example, quantitative studies on the evolution of mass loss can be problematic due to moisture absorption or difficult recovery of disintegrated material.

## 3. Polymer Composite Materials with Natural Fillers

Composites consist of two or more components, insoluble in each other, which are combined to form a useful engineering material possessing certain properties not found in the constituents [12]. Polymer composites generally consist of thermoplastic or thermoset matrix with organic (e.g., wood flour, chicken feather) or inorganic (mineral or glass materials) fillers (particulates or fibers) [13]. Thermoplastic polymers in composites usually consist of polyolefins such as polyethylene (PE), polypropylene (PP), polystyrene (PS), and polyvinyl chlorine (PVC), or of polyesters such as PLA and polyhydroxyalcanoates (PHA), of which PHB is the most common. Thermosets include acrylics, polyesters/vinyl esters, epoxies, polyurethanes, amines, furans, phenolics.

Depending on the aspect ratio, fillers can be particulates, when dimensions of three or two of the geometric axes are of a similar order of magnitude (e.g., spherical or discoid shape), or fibers, when one dimension is of several orders of magnitude larger than the other two dimensions (e.g., carbon nanotubes are about 1000 nm in length compared with only ~1 nm in diameter) [14]. This difference in aspect ratio induces differences in properties, which are enhanced with increasing fiber length (Figure 1), but also in degradation behavior, with fibers acting as continuous media that can transport aggressive factors, such as moisture, inside composite materials.

Green composites are polymer composites based on biopolymer matrix and natural fillers [16]. Wood flour and fibers are the most used natural fillers in composites. Natural fillers have lower thermal stability, therefore their use is limited to plastic materials with low melting temperatures. In fiber-reinforced composites, the matrix protects the fibers from external environmental damaging factors and transmits the externally applied loads to the reinforcement fibers, which have the ability to absorb the energy stress without deterioration of the material. Natural fibers could be of animal (hair, wool, feather, silk) or vegetal (flax, hemp, sisal, jute, kenaf, coir) origin, the vegetal ones being present in stems, leaves, or seeds of plants [17]. Vegetal fibers are themselves natural composites consisting of cellulose fibrils bounded in a matrix of hemicelluloses and lignin [18]. Lignin-rich fibers (e.g., coir: 40–45%, kenaf: ~12%) are more flexible and have higher maximum deformation (ε), while cellulose-rich fibers (e.g., cotton: 90%, pineapple leaves: 70–85%) are harder, with higher elastic modulus (E) [19].

About 2000 different plant fibers are used in various applications, including composite materials [20]. Natural fibers were introduced as replacements of inorganic ones in plastic composites because they have good mechanical (Table 1) and biochemical properties, are nontoxic, have low density, originate from renewable resources that are eco-friendly and sustainable, and are completely biodegradable, being sensitive to moisture, fungi, and insects [21]. Plant fibers are hydrophilic, thus they have low compatibility with regular plastic materials, which are highly hydrophobic. It is, therefore, necessary to increase the adhesion strength between fibers and matrix [22]. Chemical (e.g., alkaline, silane, esterification, isocyanate treatment) or physical (e.g., plasma treatment), physicochemical or biological treatments can be used for surface modification of natural fibers [23,24].

## 4. Reprocessability of Polymer Composites with Natural Fillers

Natural fibers have good mechanical properties that can be transferred to thermoplastic polymers by incorporation in composites, improving the physical, mechanical, and thermal properties of the material while decreasing its costs [27]. For example, coir fibers have high failure strain, the micro-fibrils have a helical arrangement at 45 degrees which allows stretching beyond elastic limit without rupture [28,29]. Natural fibers usually have a reinforcement effect on polymeric materials by absorbing the mechanical stress applied to the matrix; however, they tend to lower the impact strength [30].

Natural fillers improve the mechanical properties of composites, and, as a consequence, they also enhance their resistance to the action of external mechanical stress. Mechanical forces in environment are rather low and usually act as erosion and abrasion agents in combination with other aggressive factors, such as UV radiation, water, dissolved oxygen and salts, and micro biota, breaking down thermoplastic materials into litter. On the other hand, mechanical forces are much stronger in processing/reprocessing, causing, together with relatively high temperatures, changes in the structure of the properties of materials.

Some studies reported that repeated reprocessing of composite materials based on natural fibers and traditional PE, PP, and PVC plastics induces loss of tensile strength and modulus (TS and TM) and of flexural strength and modulus (FS and FM), and increase of failure strain [31,32]. According to a few other studies, the recycled composites could maintain similar mechanical properties as virgin materials, if good dispersion of fibers and strong interface adhesion between components are obtained [33,34].

Several studies reported that the dimensions of dispersed particles or fibers decrease during the first extrusion, at the contact with the molten polymer matrix, and that the equipment characteristics and processing conditions (e.g., screw rotational speed, throughput rate, barrel temperature profile), and the interfacial properties have a strong effect on the mechanical degradation of the material [35,36,37]. Strength and stiffness properties of high density polyethylene (HDPE) composites with oak wood flour (30% and 50%) and maleic anhydride (MA) (3%) coupling agent gradually decreased, while strain properties increased with successive reprocessing by extrusion and injection molding. The repeated extrusion was found to reduce the size of the dispersed wood flour and to decrease the molecular mass of the plastic matrix. The interfacial adhesion between phases was diminished, the mechanical stress was less efficiently transferred to the matrix, tensions accumulated inside the polymer matrix, decreasing the crystallinity degree and increasing the chain mobility. However, changes were relatively small after six reprocessing cycles, and the thermal stability of the material slightly increased [38]. A similar effect in decreasing the size of the dispersed phase was observed for blends of low densiy polyethylene (LDPE) with thermoplastic starch (70% starch, 30% glycerol) in a 1:1 ratio, extruded in 5 and 10 cycles before injection molding. No significant change was observed in mechanical, rheological, and dynamic mechanical properties, or in surface energy [39]. HDPE composites with 15% flax fibers and 1.5% MA-grafted polyethylene (PE-g-MA) coupling agent were extruded in 50 cycles, after which they showed better distribution of fibers in the matrix, due to the balance between the breakup of the fibers and the scission of polymer chains, with acceptable overall performance of material [40].

Various results were reported when PP was the polymer matrix in composites, depending on the type of reinforcing fiber and the presence or not of coupling agents. Flax fibers composites resisted five repeated recycles with marginal effects on mechanical properties. However, fiber breakup, induced chain scissions in the matrix, and a slight worsening of elastic modulus were observed [41]. Rice hulls or kenaf fibers (30% in composites) and PP-g-MA as the coupling agent induced marginal changes in flexural strength and thermal stability when composites were processed twice by melt mixing, kenaf fiber offering higher stability compared with rice hulls [42]. Composites with PP, lignin-based “liquid wood” and 20% hemp fiber maintained good mechanical properties and thermal stability after three reprocessing cycles [43]. Composites of extruded PP and wood flour had higher water absorption and thickness swelling after reprocessing [44].

Biocomposites of PLA and PHBV with 10%, 20%, and 30% sisal fibers became more brittle but had acceptable overall properties after two recycling steps, with only tensile strength and deformation at break decreasing after the first processing step. After the third recycling, the composite with PHBV had a slight decrease of storage modulus while the one with PLA showed significant loss of properties [45]. Oak wood fiber (50%) and PLA-g-MA as the coupling agent limited the reprocessing of PLA composites to two cycles, after which a sharp decrease in mechanical properties was observed. The high amount of wood fiber was stabilized in this case by the coupling agent. On the other hand, composites of 50% oak wood fibers with HDPE and PE-g-MA instead of PLA and PLA-g-MA accepted up to six extrusion and injection molding reprocessing cycles, with only small variation in impact properties, thermal expansion, and melt flow properties, the tensile and heat deflection being less affected [46]. Reprocessing could be possible up to three times without major loss of properties in the case of PLA/flax composites [47]. This was a good result for a polymer promoted for its biodegradability.

Kenaf fibers used alone or in a mixture with polyethylene terephthalate (PET) fibers dropped the tensile and flexural strength of polyoxymethylene (POM) composites after the first processing cycle, these parameters remained constant for another two processing cycles [48].

Knowing the reprocessability behavior helps choosing the most suitable thermoplastic matrix/natural filler (fiber) partners among various candidates, to reach the requirements of the designed new materials and of the downcycled intended products.

## 5. Biodegradability of Polymer Composites with Natural Fillers

Biodegradation rate of composite materials depends on the nature of components and on how strong they bond together, but also on the environmental conditions (e.g., temperature, moisture and pH of soil, microbial population, and nutrient supply) to which the material is subjected [49]. Biological contact occurs at the material–environment interface, therefore the area and properties of the exposed surface play significant roles, a rough surface with a high number of polar hydrophilic functional groups is much more prone to biodegradation than a smooth, hydrophobic, and inert one [50]. Natural fillers, being hydrophilic and more biodegradable, increase the adhesion of microorganisms to the composite material and favor biofouling [51]. Comparative studies on inherent characteristics of natural fibers and their performance in composites are useful in the design process of biocomposites for better choice of partners to provide special properties of the final material. Composites have a large surface area at the interface between matrix and filler. This is the weak zone of composites, which can limit their use in some applications, since the interface can act as access point of the destroying agents (biological (fungi) or chemical (moisture/oxygen)) into the rather inert plastic matrix [52]. Diffusivity of compounds through the material is also important, the amorphous domains being more biodegradable than the crystalline ones [53]. Small, localized mechanical forces could break down into smaller pieces the material that was weakened in the initial stages of degradation, increasing the exposed surface area available for further microbial attack.

Gómez and Michel Jr. [54] selected several commercially available materials proposed as “green” alternatives to classical, non-biodegradable plastics, and tested them under standardized laboratory-scale soil incubation, composting, and anaerobic digestion. They found that most natural additives do not really improve the biodegradability of recalcitrant polymers such as PE and PP, composites of copolyesters or corn-based plastics with coconut coir materials present some surface changes, and polyhydroxyalkanoate-based plastics suffer substantial biodegradation. The degradation of most bioplastics was only partial when exposed for time ranges similar to composting and anaerobic digestion recycling processes. Based on their tests, authors established relative biodegradability orders of materials, in which conventional plastics with or without additives have the lowest biodegradability, while the best biodegradability was for PHA exposed for 660 days to incubation in soil and for plastarch (biodegradable thermoplastic starch-based resin) exposed for 50 days in anaerobic digestion and for 115 days in compost. This study considered only the natural materials used as additives or components in blends, with no interest in the natural materials used as fillers or fibers in composites.

Wood plastic composites (WPC) are materials that usually contain 30–55% polymer matrix (PP, PE, PVC), 30–70% wood particles (softwood/hardwood flour or shavings), and 0.5–15% additives, and are designed for long-term performance, shape flexibility, good stiffness and working properties [55,56]. Classic, fossil-based plastics are biologically inert materials, but this property is only partly transferred to WPC because the plastic matrix cannot totally encapsulate the wood particles, which can absorb moisture to levels that makes them susceptible to fungal attack [51]. While microbiological attack is less of a problem for most indoor applications, it is of great importance for the durability of WPC with outdoor uses. Fungi, especially white rots, can rapidly colonize and decay WPC, favored by warm temperatures and high moisture environments [57].

Candelier et al. [58] reported that WPC of BIOPLAST GS2189 biopolymer (from PLA and potato starch) and spruce sawdust have good resistance to fungal and termite attack (at 27 °C and 70% relative humidity—RH), for exposure times of 2 and 4 weeks, respectively. The resistance decreased with increasing wood content, so at 30% wood the material suffered slight termite attacks.

Schirp and Wolcott [57] found that determination of mass loss is a more sensitive indicator of fungal decay than strength and stiffness measurements for WPC of HDPE and maple wood flour that was incubated for three months with wood decay fungi. Deterioration by mold fungi of LDPE composites with 30% various biomass fillers was strongly influenced by the aspect and composition of fillers. The particles with higher length-to-diameter ratio induced biofouling by increasing the surface area accessible to fungal attack, while fillers containing soluble or easy hydrolysable fractions (milled straw of seed flax and hydrolyzed keratin of bird feathers), enhanced biodegradation, providing an easily available source of carbon for micromycetes [59]. Polymer composites with 50% content of hazelnut husk flour showed better durability over 16 weeks under the attack of white or brown root fungus when HDPE was the polymer matrix instead of PP, and when PE-g-MA was used as the compatibilizing agent to stabilize the interface [60]. The biodegradation of LDPE/alkali treated corn flour composites exposed to soil burial for 6 months increased with the amount of biomass component [61].

PLA is a synthetic thermoplastic aliphatic polyester that is completely derived from renewable resources, such as corn starch or sugar cane, and has similar characteristics to classical PP, PE, and PS but is considered biodegradable [62]. While of natural origin, PLA shares the common hydrophobicity of the fossil-based equivalent polyolefins. It has poor surface adhesion with natural fibers or fillers, which usually have high polar character, and requires compatibilizing agents to stabilize the interface in composites [63,64]. Chitosan, especially in high amounts, increases the hydrophilicity of PLA composites, favoring moisture uptake and the attack of microorganisms in soil burial for 150 days. Tested sheets had small mass loss, but significant embrittlement and changes in mechanical, thermal, and surface properties were observed [65]. The usual temperature in the environment, soil or compost, is below the glass transition (Tg) of PLA (60 °C) [66,67]. The rigidity of macromolecular chains slows down the migration of moisture and microbial agents into the material, decreasing its degradation rate [68,69]. Using plasticizers or blending PLA with polymers that are more flexible can decrease the glass transition, favoring biodegradation [70,71]. A crystallinity increase was observed in chitosan loaded PLA films exposed to white root fungus, which mainly attack the amorphous regions, having higher flexibility of macromolecular chains [72]. PLA degradation starts by abiotic chemical hydrolysis and continues by enzymatic hydrolysis under microbial attack, which occurs preferably in the amorphous regions, and by biotic assimilation of degradation products [73,74]. Careful tuning is needed to balance the necessity of maintaining suitable properties of a material during the entire service life in environmental conditions and the need of rapid degradation when the material becomes waste.

The biodegradation rate of PLA under controlled composting conditions is increased by hydrophilic fillers such as starch or kenaf bast fibers [75,76,77]. Fiber length and orientation has a significant effect on the mechanical properties, higher stiffness and tensile strength being observed in the direction of unidirectional oriented fibers [78]. This also enhances the biodegradability of flax/PLA composites in compost, by acting as transporting channels for humidity and microorganisms that thus have access to the inside of material. The initial material (I in Figure 2a) start to swell, the contact interface is enlarged, cracks appear (II in Figure 2a) and grow with enzymatic degradation of the flax fiber (Figure 2b), and by hydrolytic degradation of the PLA matrix [79] (III in Figure 2a), eventually the material becomes brittle, with loss of debris (IV in Figure 2a). The biodegradability in soil burial of flax fiber reinforced PLA composites also depends on the presence of amphiphilic additives used as accelerators for biodegradation of PLA, the mandelic acid leading to the removal of PLA from the surface of material and to 20–25% mass loss after 50–60 days, while for dicumyl peroxide, only 5–10% mass loss was observed after 80–90 days [80].

For the composites of PLA/thermoplastic starch in a 3:1 ratio as matrix and 30% coir as filler, biodegradation in controlled composting conditions was strongly influenced by the starch component, which was totally biodegraded after the incubation period, while the fibers had only a marginal role in the process. The use of 1% MA as a compatibilizing agent slightly decreased the bioactivity by stabilizing the interface. Biofilm appeared on biodegraded materials, due to bacterial and fungal fixation and growth [82].

PHB is a polyhydroxyalcanoate bio-based polyester produced through intracellular synthesis by soil bacteria and archaea. It has similar properties as classical, fossil-based PP, but with the advantage of a green origin and biodegradability. PHB is morphologically organized in spherulites of partly ordered lamellae consisting of crystalline and amorphous domains in alternating layers. Processing conditions strongly affect the size of organized domains in PHB, and thus influence the biodegradability [83,84]. Various biodegradation rates in compost were reported for PHB, depending on material type, from total degradation in 30 days for thin films, down to only ~4% mass loss after 150 days for injection-molded dog bones [85,86]. Biodegradation of PHB films when composted under accelerated aerobic test conditions was strongly improved by bacterial cellulose (BC), reaching 80% after 30 days for composites with 10% BC instead of 50 days for PHB alone. Biodegradation occurred mainly in the amorphous regions of PHB, leading to morphology changes in semi-crystalline lamellae and to increased crystallinity degree [53].

Poly(butylene succinate) (PBS) is a thermoplastic aliphatic semicrystalline polyester that has properties comparable to PP and is also biodegradable. Hemp fillers in amounts of up to 70% in composites, accelerate the biodegradation of PBS in enzymatic hydrolysis, and especially in soil burial, shives having higher effects than fibers [87].

Nanoclays consist of layered mineral silicate nanoparticles with large specific surface and high surface energy, which create strong interfacial interactions with the polymer matrix in composites, depending on the degree of dispersion and the shape and size of particles or aggregates [88,89]. Montmorillonite nanoclays are hydrophilic in nature but they have good bonding with PP, which stabilizes the interface in composite materials containing PP as a matrix and natural fibers (kenaf, coir, or wood) as fillers. This retards the biodegradability by decreasing the access of moisture to the inside of material, thus lowering the water absorption in early stages of soil burial, and by localizing inside clay, the water is absorbed in higher amounts at later stages of biodegradation [90]. Montmorillonites hindered the enzymatic attack of α-amylase on polyvinyl alcohol (PVA)/starch composites, with nanocore having higher effects than bentonite and Peruvian clay. Dissolution and erosion were the main occurring processes, which were accompanied by enzymatic degradation of the starch phase in composites [91].

Composites with poly (butylene-adipate-terephthalate) ternary copolymer and starch as the matrix and 10–30% rice husk as the filler, tested in simulated ground, showed that biodegradation of rice husk filler was hindered by the polymer matrix phase, in which the microbial attack preferentially occurred [92].

Moreira et al. [93] developed special biodegradable composite materials that improve microbial activity during biodegradation by burial for up to 35 days in a commercial substrate for seedling production. They prepared composites that contain 25% PVA, 20% starch, 20% different natural fillers (sugarcane bagasse, oat hulls, or silkworm exuvia), and various nutrients to stimulate the biodegrading microbiota. Sugarcane bagasse as a filler was preferentially attacked by bacteria, and induced the lowest biodegradation rate, water absorption capacity, and water solubility, the composite being suitable for long-term application. Oat hulls and silkworm exuvia were predominantly attacked by filamentous fungi and stimulated the microbial activity in soil.

Gamma irradiation in doses up to 500 kGy improved the biodegradability of the blends obtained from polypropylene (PP)/high-melt-strength polypropylene (HMSPP) structural foams and sugarcane bagasse or PLA, through surface erosion of the material, which increases the water penetration, thus favoring the attack of microorganisms [94,95]. Gamma irradiation also induced biodegradability in composites of bio-inert PP with biomass fillers (*Eucalyptus globulus*, pine cones, *Brassica rapa*) under the attack of white rot fungus for 7 weeks. Rougher texture was observed for the outer surface of tested specimens, while cracks and scrap particles were observed over the entire matrix surface at the interface with biomass fillers [96].

## 6. Weathering and Accelerated Ageing of Polymer Composites with Natural Fillers

Under environmental factors such as sunlight, temperature, humidity, oxygen, and mechanical stress, plastic materials lose physical integrity and continuously break down into smaller particles, known as microplastics [97]. Delamination without evident visual fragmentation might also be a mechanism of microplastics formation in salt marshes [98]. It was estimated that ~6.2 Mt macroplastics and ~3.0 Mt microplastics were lost in the environment in 2015 [99]. Due to their very small size (below 5 mm) and low density, microplastics are easily transported over very long distances by wind and water [100,101,102], becoming pervasive in various environments, from urban zones [103] to protected natural areas [104]. Microplastics generate pollution in agroecosystems [105], affect living species, including humans, by ingestion or inhalation [106], the ecotoxicity and environmental impact being strongly related to their physical and chemical characteristics [107], and can act as vector for potentially pathogenic bacteria [108] and for toxic trace-element uptake by aquatic and terrestrial organisms [109]. Microplastics ingested by organisms can be transferred between different trophic levels [110]. Nanoplastics (smaller than 100 nm in one direction) have large specific surface area and surface functional groups resulted from environmental degradation and can be more hazardous than microplastics due to their ability to permeate biological membranes [111]. Biodegradable polymers could decrease the amount of plastic debris; however, they cannot significantly diminish the marine litter, since the marine environment is different from the terrestrial one, for which the environmental polymers are usually designed [112].

In 2015, Shahzad and Isaac [113] published a comprehensive report on weathering studies found in literature at that time on lignocellulosic polymer composites, with detailed discussions on the UV and moisture effects on composites components (fiber and polymer matrix), methods to improve their resistance, and testing methods to evaluate weathering properties. Therefore, the focus here is mainly on studies reported after the year 2015.

Harsh environmental conditions such as high humidity or moisture, temperature, or strong UV radiation induce the weakening of materials, with negative effects on their utilization. Investigations are necessary to determine fatigue resistance in environmental conditions and the evolution rate of degradation processes. Studies on natural ageing are time consuming, being suitable for materials with short-term use; however, they cannot be used for lifetime prediction of materials intended for long-term use in environmental conditions. Artificial aging tests are designed to accelerate the ageing processes by simulating the outdoor climate in laboratory conditions. For example, UV lamps are used to mimic natural sunlight in accelerated photo-oxidative degradation, and heating at relatively high temperatures in forced air circulation is used to accelerate thermo-oxidative ageing. There is, however, the concern that the results obtained from artificial conditions cannot be entirely extrapolated to practical applications [114]. Friedrich [115] comparatively revised artificial and natural weathering results for wood–polymer composites used in cladding applications and found comparable effects in the material. However, these effects needed a 7.4 times longer exposure period under natural conditions compared with accelerated ones. Special algorithms were able to determine the decline in strength of the material as function of exposure time and fiber content [116]. Azwa et al. [117] found that exposure to laboratory accelerated ageing for about 400–2000 h corresponds to at least 2 years exposure to natural weathering. Using principal component analysis (PCA), a statistical method that considers all degradation factors and their effect at the same time, Badji et al. [118] found correlations between accelerated artificial ageing and outdoor natural weathering of hemp fiber reinforced PP composites. Based on this, they determined acceleration factors for different materials, as a ratio between the artificial ageing time and the natural weathering time needed to obtain the same effect. For example, after 250 h of exposure to accelerated ageing, neat PP polymer suffered similar deterioration as in 1 year of weathering, while up to 750 h of ageing were necessary for PP composites to obtain equivalent correlations.

Ageing and weathering induce negative effects at the surface of materials, which can further spread inside material, usually up to depths of about 0.5 mm. This is limited by shallow penetration of UV radiation in polymeric materials and by a low diffusion rate in solid materials in absence of moisture (or other liquid) that can transport attacking factors inside the material and degradation products outside it.

In environmental conditions, solar radiation acts in combination with air as an oxidizing agent, to start chemical reactions that cause changes in color (e.g., yellowing or discoloration) and appearance of materials, which can continue up to embrittlement and loss of physical integrity [119]. UV radiation (100–400 nm) represents only about 6.8% of the spectral range of solar radiation (100–3000 nm) but it has sufficiently high energy to break chemical bonds (Table 2), thus damaging the outdoor exposed materials. The UV-B radiation (315–280 nm) is the shortest wavelength solar radiation reaching the Earth’s surface (UV-C radiation below 280 nm is totally absorbed by the ozone layer) and is the most aggressive to polymeric materials [113].

Photodegradation of polyolefins involves Norrish reactions of ketones and aldehydes, the type I ones leading to free radicals generating cross-linkings or chain scissions, and the type II ones leading to carbonyl and vinyl groups (Figure 3).

The presence of chromophores in a material enhances the absorption of UV radiation, generating radical species that initiate degradation, while the presence of photo stabilizers retards photodegradation by inactivation of formed radicals. The balance between the concentration and reactivity of chromophores and of photo stabilizers dictates the rate of photodegradation. It has to be considered that the concentration of active photo stabilizers continuously decreases during prolonged exposure to UV radiation, while more chromophores are formed as degradation products, so that after a certain stage (induction period), the photo-oxidative degradation has autocatalytic evolution.

Lignin and extractives (mainly the phenolics) in natural fillers are very sensitive chromophores in the UV region, initiating the degradation processes with formation of new chromophore functional groups such as carboxylic acids, quinones, and hydroperoxy radicals that induce the yellowing aspect of photodegraded wood in wood-based polymer composites [120,121]. Eventually, loss of color occurs, and the material becomes grey. Gaseous compounds such as methanol, CO, and CO_2_ are formed under the action of UV radiation from functional groups such as methoxyl, carbonyl, or carboxyl [122]. PP is very sensitive to UV radiation that induces photo-oxidative degradation in the presence of oxygen from air, with the formation of oxygen-containing chemical groups that act as chromophores, absorbing the UV radiation, thus accelerating the photo-oxidative processes [123,124]. The combined effect of wood fiber and PP matrix doubles the effect of degradation, when filler content increases from 25 to 50% [125].

Since its introduction by Mellor in 1973 [126], the Carbonyl Index, which refers to the IR absorbance of the carbonyl groups (νC=O) at 1712 cm^−1^, was extensively used to monitor the photo-oxidation of polymeric materials. However, some reports indicate that this is not a sensitive approach to determine the initial stages of the photo-oxidation processes of bio-based materials, because the formation of carbonyl groups is usually detected by IR only after about 40 h of photo-oxidative attack, and the maximum of the absorption band gradually shifts from dimer acids to esters after 80 h of exposure, which makes it difficult to analyze the C=O band in the 1500–1900 cm^−1^ range of the IR spectrum. The methyl band at 1456 cm^−1^ was proposed instead, as a more suitable indicator that correlates well with changes in other parameters, such as crystallinity degree, molecular mass, or micro-hardness [127]. Indeed, the C=O bond in other functional groups, which might be already present in the system, also has IR absorbance in this region, for example at 1712 cm^−1^ for carboxylic acids in dimer form [128], at 1735 cm^−1^ for esters [129], and at 1780 cm^−1^ for γ-lactones [130], and interferences might occur with the formation of carbonyl groups as a result of photodegradation.

Moisture absorption and water intake under environmental exposure are the main factors whose evolution can significantly drop the mechanical properties of composites [131,132]. Natural fibers have high affinity towards moisture (Table 3) and might swell during ageing or weathering, the dimensional changes leading to swelling stress, with the formation of cracks that can advance up to integrity failure [133]. On the other hand, moisture entering the material through fibers acts as plasticizer of polymer matrix, weakening the interfacial bond with the filler, which, again, can generate and propagate cracks [134,135]. Ageing of hemp containing composites of PLA and polyesters showed that moisture absorption is strongly related to the volume of fiber, volume of voids, and cellulose content in materials. Pretreatment can induce irregularities on the surface of the fibers, leading to higher water uptake, as reported for mercerised wood fiber/PP composites [136]. Increasing length and content of coir fibers in epoxy composites favored water uptake [137]. In contrast, smaller lengths allow fibers to pack closely and improve fiber adhesion, with less and smaller available free spaces. Moisture absorption in pultruded jute/glass fiber hybrid composites was found to be non-Fickian [138].

Ageing of composites exposed in environmental conditions is an effect of an aged interface between the polymer matrix and the reinforcing filler under the action of degradation agents that can penetrate the material, usually through fibers. Nguyen-Duy et al. [139] analyzed single fiber micro-composites of PP and hemp by pull-out tests on an in situ micro-tensile machine and found that the fiber–matrix adhesion was severely weakened during moisture accelerated aging, allowing fibers to be pulled out easily from the matrix.

Swelling/shrinking of anisotrope natural fillers occurs inside thermoplastic matrices at water absorption/desorption cycles. Internal tensions appear due to the difference between reversible dimensional changes of fillers and plastic deformation of the matrix, leading to stress-induced damages of the filler/matrix interface, and pull-out of fillers, with the formation of voids [140]. Fillers occupy the available voids in subsequent swelling, increasing the interfacial tension. Eventually, after repeated absorption/desorption cycles under the combined effect of moisture and temperature, gaps appear around filler particles, leading to cracks inside material that extend with exposure.

Immersion in water or other fluids was considered as extreme conditions to observe ageing of interface [141]. Organic solvents (e.g., petroleum-based liquids, vegetable oils) can enter the surface voids of composites, weakening the interface and swelling the hydrophobic matrix, resulting in decreased tensile strength of the material. Water exposure generates fiber pull-out from composites; hydrolysis reactions degrading the biomass material generate high amounts of active species that can initiate the degradation of the polyolefin matrix (e.g., PP), leading to decreased molecular mass [142,143]. Fiber pull-out leading to material failure was reported during weathering in various conditions of bagasse fibers reinforced polymer composites [144].

Degradation during weathering occurs firstly in the amorphous regions, leading to relative increase of the crystalline phase. New crystalline domains appear by the rearrangement of shorter chains resulting from the degradation of macromolecules in the amorphous phase. Advanced weathering eventually also affects the crystalline phase [145].

Composites of PP with natural fibers (hemp, jute, sisal) showed decreased tensile strength after 3 months exposure in drinking water, peanut edible oil, petrol, industrial liquid waste, or alkali solution, with the highest variation for polar alkali (5% NaOH) and lowest variation for non-polar peanut oil. The observed increased mass was due to liquid absorbance, with maximum gain in non-polar petroleum liquid [146]. Catto et al. [147] studied WPC with PP-ethylene vinyl acetate (EVA) copolymer obtained from post-consumer waste cap (PP) and its internal liner (EVA), as a plastic matrix, and wood flour (eucalyptus or pine) as the dispersed phase, in the presence or absence of PP-g-MA coupling agent. The composites were weathered 90 days in natural conditions of sun, rain, and high temperatures, then analyzed by respiratory tests and by fungal decay tests against four white rot fungi for 12 weeks. Natural weathering induced micro and macro cracks on the surface of composites, making them more susceptible to biodegradation. However, even after 90 days the materials could be described as resistant or very resistant to fungal attack. Biodegradation was more advanced in the presence of pine wood flour instead of eucalyptus, when the coupling agent was used, and in the presence of *Fuscoporia ferrea* fungus, which was more effective in surface colonization. Outdoor weathering for 270 days induced surface degradation by stress cracking, resulting in micro and macro cracks, which facilitates subsequent biodegradation [148].

PP and PS are more suitable than PE as matrix in wood plastic composites with good resistance to environmental (UV and humidity) and mechanical stress [149]. For the composites of PP and wood dust from Babool, Sheesham, Mango, and Mahogany trees, weathering in water was more advanced than in soil and increased with wood content. Sheesham dust induced the highest resistance to biodegradation. Natural weathering and biodegradation were related, the composites with a less stable interface being both more biodegradable and more susceptible to weathering [150,151]. Prolonged exposure of PP composites with birch plywood sanding wood in an accelerated weathering chamber for 1032 h led to rougher surfaces with microcracks, faded appearance, change of gloss, whiteness, and decreased microhardness. Recrystallization occurred in the surface layers of PP, decreasing its deformation ability and increasing the tensile modulus [152]. Differences in thermal behavior (e.g., changes in characteristic temperatures and in kinetics of the mass loss) were observed by thermogravimetric analysis (TG) for PP/starch blends (70/30 wt %) exposed for a period of 4 years to natural weathering that induced a loss of stabilizers and subsequent photodegradation [153]. Wood flour/PVC composites containing an antifungal agent lost flexural properties during UV-accelerated weathering aging and natural weathering, the effect being stronger under UV irradiation [154]. Thermoplastic starch obtained from agricultural waste imparted better resistance to natural weathering of PP blends for 6 months compared with starch of regular origin. Surface cracking and the presence of microorganisms was observed, accompanied by deterioration of mass, tensile properties, thermal properties, and relative molecular mass and by the increase of carbonyl index, under a combined effect of UV radiation, oxidation, moisture, temperature, and microbial attack [155].

Wang and Petrů found that surface treatment (acetylation, alkalization, silanization, alkali-silanization) of flax fibers improves water resistance and damping properties of fiber reinforced polymer composites, acetylation treatment having the best results in stabilizing the material. Freeze–thaw cycles accelerated the aging effect, increasing moisture uptake with subsequent alteration of interactions at the matrix/reinforcement interface, and decreasing flexural properties [156]. Enzymatic treatment of cellulose fibers helped to preserve mechanical and thermal properties, as well as surface chemistry, while losing rheological properties in PLA reinforced composites [157]. Compared with NaOH-treated jute fibers, microcrystalline cellulose induced better resistance to UV degradation (above 290 nm) as well as biodisintegration under biotic and abiotic attack in soil burial of composites with an ethylene-propylene copolymer [158]. Epoxy composites with pineapple leaves, coir fibers, or their mixture lose mechanical strength at an accelerated rate after 30 days of soil burial, biodegradation increasing with the content of pineapple leaves, which suffered substantial hydrolysis. NaOH treatment increased the biodegradability of coir in soil burial for 110 days but stabilized the pineapple leaves by removal of polar groups (e.g., –OH) [159]. Chemical treatment with 10% sodium bicarbonate solution of flax fibers used as reinforcing agents in epoxy based composites improved interfacial adhesion between the fibers and matrix. The flexural properties of fibers are retained, and composites had improved resistance against ageing in marine environment simulated conditions (salt-fog spray up to 60 days). However, this pre-treatment slightly destabilizes the interface in the case of jute fibers, worsening the durability of corresponding composites [160].

Plastic materials are widely used in agriculture as mulch films, greenhouse covers, bale covers, silage bags, containers, etc. [161]. Traditional plastics (e.g., HDPE, LDPE) have over 70 years of use in plasticulture (see [162] for an extensive historical review), their use as mulch films being very useful to maintain constant temperature and moisture in soil and good weed management; however, they can decrease soil fertility if degraded mulch fragments remain and accumulate in soil [163]. This can be avoided by using biodegradable mulch films. Plasticized PLA/PHB with 3% carbon black composite films, tested for 90 days as mulch film, maintained their integrity, with no visible cracks or defects, but had a 15–25% decrease of average molecular mass, decline in PHB content, and increase of storage modulus. These effects are stronger for the parts of films that were exposed to sunlight compared with those in shadow of the plants or buried in soil [164]. When wood flour was added in low amounts (25%) to PHBV, PLA, or PE, wood filler was covered and protected by the polymer matrix, preserving the mechanical properties of the composites over 12 months of outdoor exposure; however, weathering dropped the properties for a high content (50%) of wood flour, when mold growing on the shaded surface was also observed [165].

Kenaf fibers induced surface damage, increased carbonyl index, crystallinity of linear low density polyethylene (LLDPE) and mass loss, as well as decreased tensile properties in LLDPE/PHA/kenaf composites weathered for 6 months, the effects increasing with fiber load and exposure period [166]. Ariawan et al. [167] studied the natural weathering for 12 months of polymer composites containing unsaturated polyester resin (Reversol P9565) and kenaf fiber nonwoven mat pretreated by alkali (6% NaOH, 3 h) or heat (140 °C, 10 h, circulating air) to improve tensile and crystallinity index. Materials gradually changed colors to lighter nuances and showed increasing content of carbonyl and vinyl groups as a result of photodegradation. Surface deterioration and fiber damage were observed, which led to a significant decrease of mechanical properties. The alkali pretreatment induced better environmental resistance compared with the heat treatment. Composites of commercial thermoplastic starch and ester components blend (Mater-Bi KE03B1) with kenaf and cotton fibers show strong interactions at the matrix–fiber interface, which limits water uptake in hydrothermal ageing. The fibers hindered the photo-oxidation of starch in the matrix. Kenaf fibers, with more lignin content than cotton ones, induced lower stability of composites under artificial photo-oxidation (with 280–700 nm radiation) or soil burial (over 330 days), and especially under a synergistic effect of both degrading agents [168].

Coriander straw fibers used as reinforcing agents for PP and bio-based LDPE in thermoplastic composites preserves mechanical performances after accelerated UV and/or hydrothermal ageing, as well as after five reprocessing cycles [169].

## 7. Conclusions

Degradation of plastic composites under the action of aggressive environmental factors is a subject of great interest to the scientific community. In 2016, Meng and Wang [170] reviewed the studies on aging of fiber reinforced polymer composites and pointed out that the processes occurring at the level of microscopic structures should be better studied, for deeper understanding of degradation mechanisms, especially at initial stages, which are difficult to evaluate at a macroscale. This observation is still valid at the end of 2019, when only few reports (e.g., the work of Nguyen-Duy et al. [139]) were found to study the matrix/fiber interface at micro-scale and its deterioration during ageing. The advance of new analysis and characterization techniques, such as high-resolution TEM (transmission electron microscopy) and depth-profiling XPS (X-ray photoelectron spectroscopy), which can provide information on the morphology and atomic composition beneath 1 to 10 molecular layers that can be evaluated through surface-sensitive methods, or nanoindentation and microscratch tests, are able to determine micromechanical properties of surfaces and can open new perspectives.

## Figures and Tables

**Figure 1 polymers-12-00166-f001:**
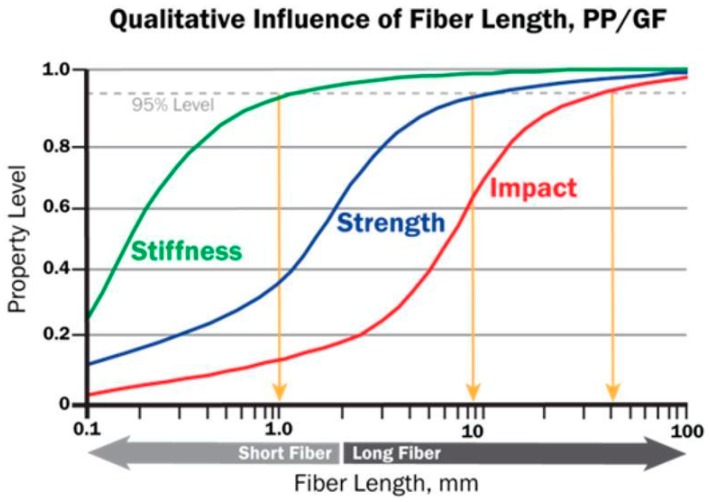
Dependence of properties with fiber length in glass fiber reinforced polypropylene (PP) Reprinted with permission from [15].

**Figure 2 polymers-12-00166-f002:**
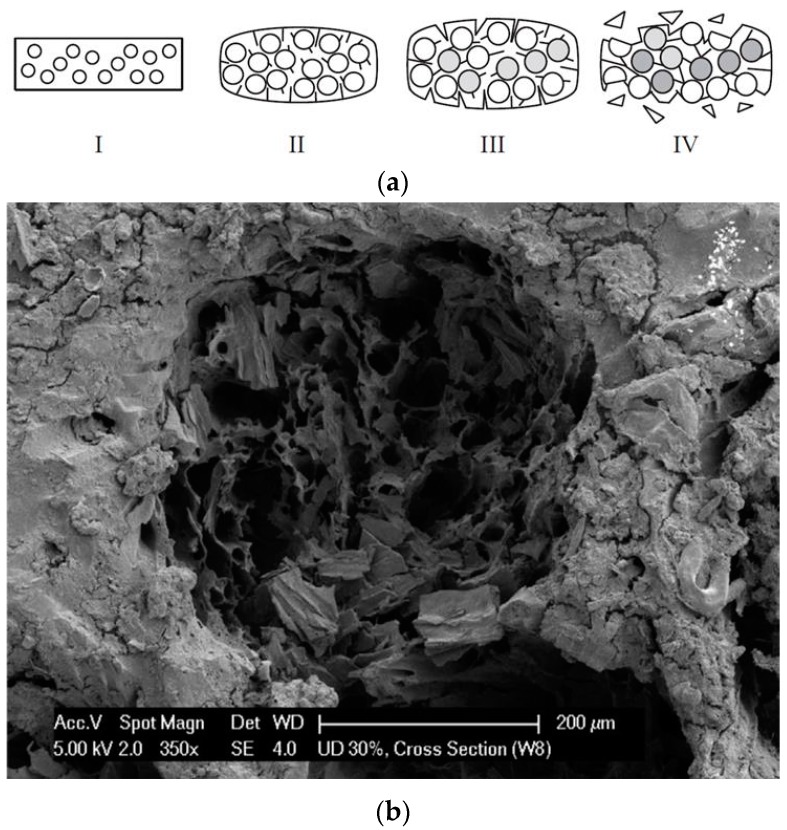
Degradation states (**a**) and hole remaining after the degradation of the flax fiber (**b**) in polylactic acid (PLA)/flax composites. Reprinted with permission from [81].

**Figure 3 polymers-12-00166-f003:**
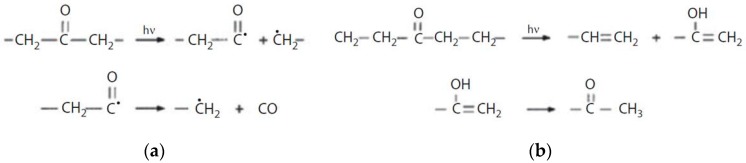
Norrish I (**a**) and Norrish II (**b**) reactions occurring in photodegradation of polyethylene Reprinted with permission from [113].

**Table 1 polymers-12-00166-t001:** Mechanical properties of several natural and synthetic fibers, data from [24,25,26].

Fiber	Density (g/cm^3^)	Elongation at Break (%)	Elastic (Young) Modulus (GPa)	Tensile Strength (MPa)
Aramid	1.4	3.3–3.7	63–67	3000–3150
Carbon	1.4	1.4–1.8	230–240	4000
E-glass	2.5	0.5	70	2000–3000
S-glass	2.5	2.8	86	4570
Polyester	1.2–1.5	2.0–4.5	2	40–90
Polyhydroxyalkanoates	1.1–1.4	1–6	3–6	35–100
Cotton	1.2–1.6	7.0–8.0	5.5–12.6	250–500
Coir	1.2	24–51	6 (40)	140–593
Flax	1.2–2.4	2.3–3.2	27.6–80.0	500–1500
Hemp	1.3	2–40	45 (70)	690 (530–1100)
Jute	1.2–1.8	1.5–2.5	10–55	325–800
Kenaf	1.2–1.6	1.6	41 (53)	745–930
Sisal	1.2-1.5	2.0–3.2 (8)	9.4–22.0	310–855
Abaca	1.5	3.4	41	410–810
Henequen	1.4	4.8	13.2	500
Pineapple	1.5	0.8–3.2	82	1020–1600
Banana	1.3	2.0–3.7	27–32	720–910
Nettle	1.5	1.7	38	650
Ramie	1.4	1.2–3.7	23–44	500–915

**Table 2 polymers-12-00166-t002:** Dissociation energy of several bonds and the corresponding radiation wavelengths. Reprinted with permission from [113].

Bond	Bond Dissociation Energy (kJ/mole)	Wavelength (nm)
C–C (aromatic)	519	231
C–H (aromatic)	431	278
C–H (methane)	427	280
O–H (methanol)	419	286
C–O (ethanol)	385	311
C–O (methanol)	373	321
CH_3_COO–C (methyl esters)	360	333
C–C (ethane)	352	340
C–Cl (methyl chloride)	343	349
C–COOCH_3_ (acetone)	331	362
C–O (methyl ether)	318	376

**Table 3 polymers-12-00166-t003:** Equilibrium moisture content (EMC) of several natural fibers at 21 °C and 65% RH. Reprinted with permission from [133].

Fiber	Wood	Jute	Flax	Hemp	Ramie	Sisal	Pineapple
**EMC (%)**	12	12	7	9	9	11	13

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
