# Peer review of "Environmental Degradation of Plastic Composites with Natural Fillers—A Review"

_polymers, 2020, doi:10.3390/polym12010166_

Round 1

Reviewer 1 Report

The manuscript under consideration deals with environmental degradation of plastic composites containing natural fillers. It is well written contribution suitable for publication after minor revision where author should address the problem of microplastics formation due to erosion of non-degradable plastic components of such composites.

Author Response

Rev1: The manuscript under consideration deals with environmental degradation of plastic composites containing natural fillers. It is well written contribution suitable for publication after minor revision where author should address the problem of microplastics formation due to erosion of non-degradable plastic components of such composites.

Author thanks reviewer for valuable suggestion. Discussion was added in lines 352-366 on the problem of microplastics, as following:

Under environmental factors such as sunlight, temperature, humidity, oxygen and mechanical stress, plastic materials lose physical integrity and continuously break down into smaller particles, known as microplastics [97]. Delamination without evident visual fragmentation might be also a mechanism of microplastics formation in salt marshes [98]. It was estimated that ~ 6.2 Mt macroplastics and ~3.0 Mt microplastics were lost in environment in 2015 [99]. Due to their very small size (below 5 mm) and low density, microplastics are easily transported over very long distances by wind and water [100-102], becoming pervasive in various environments, from urban zones [103] to protected natural areas [104]. Microplastics generate pollution in agroechosystems [105], affect living species, including humans, by ingestion or inhalation [106], the ecotoxicity and environmental impact being strongly related to their physical and chemical characteristics [107], and can act as vector for potentially pathogenic bacteria [108] and for toxic trace-element uptake by aquatic and terrestrial organisms [109]. Microplastics ingested by organisms can be transferred between different trophic levels [110]. Nanoplastics (smaller than 100 nm in one direction) have large specific surface area and surface functional groups resulted from environmental degradation, and can be more hazardous than microplastics due to their ability to permeate biological membranes [111].

Reviewer 2 Report

Comments on: Environmental degradation of plastic composites with natural fillers a review

The topic is very interesting, current and worth of publication, but I have some comments that the author need to consider.

ALL THE PAPER: English needs to be improved. I suggest a check from a native speaker. Moreover, in the manuscript there are some logical jumps and changes of arguments. I suggest a check from this point of view. Another important point is about the distribution of the references in the text. Often the author makes statements without the presence of adequate citations. This is particularly strange in a review article. Therefore I ask the author to insert these citations.

INTRODUCTION:

I suggest to the author to add a brief scheme of the article in the introduction to guide the reader.

PAG 2 LINE 77- The title of the paragraph refers generically to natural fillers, but in the paragraph the author speaks mainly about of natural fibers and in Tab. 1 the particles (e.g wood sawdust, dried fruit powder, etc.) are totally absent. I suggest the author to classify and define better the difference between particles and fibers because the difference in terms of degradation and properties is very significant. PAG 3 LINE 113 - Since the topic of the review is environmental degradation, I suggest to justify the presence of a paragraph concerning reprocessing. I also suggest to the author to add, if it is possible, some conclusive consideration about the most favorable coupling between polymer and fibers in terms of reprocessing PAG 8 line 324 - I ask the author to discuss better the degradation mechanisms that are involved during weathering and agening. There are some articles that try to explain how these happen. For example

Fortini A., Mazzanti V. Combined effect of water uptake and temperature on wood polymer composites, Journal of applied polymer science 135(35),46674, 2018

Badji, C., Beigbeder, J., Garay, H.Email Author, Bergeret, A., Bénézet, J.-C., Desauziers, V. Natural weathering of hemp fibers reinforced polypropylene biocomposites: Relationships between visual and surface aspects, mechanical properties and microstructure based on statistical approach, Composites Science and Technology Volume 167, 20 October 2018, Pages 440-447

Furthermore, it could be convenient for the reader to have a table with the references in which it is possible to quantify the trend of the mechanical properties as a function of degradation

Author Response

Rev2: The topic is very interesting, current and worth of publication, but I have some comments that the author need to consider.

Author thanks reviewer for valuable suggestions.

Q1: ALL THE PAPER: English needs to be improved. I suggest a check from a native speaker. Moreover, in the manuscript there are some logical jumps and changes of arguments. I suggest a check from this point of view. Another important point is about the distribution of the references in the text. Often the author makes statements without the presence of adequate citations. This is particularly strange in a review article. Therefore I ask the author to insert these citations.

A1: Text was carefully checked, English was improved, citations were added when necessary.

INTRODUCTION:

Q1: I suggest to the author to add a brief scheme of the article in the introduction to guide the reader.

A1: This was added in lines 34-43, as following:

Here we present an overview of the latest reports on the effect of most usual environmental factors on the degradation of plastic composites with natural fillers. We firstly introduce some general aspects on environmental degradation of materials, defining the terms and testing practices. Then we present some generalities on polymer matrices and on natural fillers used in composites. The alteration of mechanical properties under the effect of repeated processing is described, with particular examples. Biodegradability is discussed, with examples of composites with bio-inert or with biodegradable polymeric matrices, emphasizing the role of matrix/filler interface and describing different degradation stages. Effect of photoradiation and of moisture was discussed, with particular examples, in the section dedicated to weathering and accelerated ageing of polymer composites with natural fillers.

Q2: PAG 2 LINE 77- The title of the paragraph refers generically to natural fillers, but in the paragraph the author speaks mainly about of natural fibers and in Tab. 1 the particles (e.g wood sawdust, dried fruit powder, etc.) are totally absent. I suggest the author to classify and define better the difference between particles and fibers because the difference in terms of degradation and properties is very significant.

A2: Difference between particles and fibers was defined in lines 99-108, as following:

Depending on the aspect ratio, fillers can be particulates, when dimensions on three or two of the geometric axes are of similar order of magnitude (e.g. spherical or discoid shape, respectively), or fibers, when one dimension is of several orders of magnitude much higher than the other two dimensions (e.g. carbon nanotubes are about 1 000 nm in length compared with only ~ 1 nm in diameter) [14]. This difference in aspect ratio induces differences in properties, which are enhanced with increasing fiber length – Figure 1, but also in degradation behaviour, fibers acting as continuous media that can transport aggressive factors, such as moisture, inside composite materials.

Figure 1. Dependence of properties with fiber length in glass fiber reinforced PP [15].

Particles are not present in Table 1 since they do not possess the necessary physical integrity to determine the respective mechanical properties at macroscale.

Q3: PAG 3 LINE 113 - Since the topic of the review is environmental degradation, I suggest to justify the presence of a paragraph concerning reprocessing. I also suggest to the author to add, if it is possible, some conclusive consideration about the most favorable coupling between polymer and fibers in terms of reprocessing

A3: Paragraphs were added, in lines 140-146 and 199-201 to introduce and to conclude the presence of reprocessing paragraph in the paper, as following:

Lines 140-146: Natural fillers improve the mechanical properties of composites, and, as consequence, they also enhance their resistance to the action of external mechanical stress. Mechanical forces in environment are rather low and usually act as erosion and abrasion agents in combination with other aggressive factors, such as UV radiation, water, dissolved oxygen and salts, and micro biota, breaking down thermoplastic materials into litter. On the other hand, mechanical forces are much stronger in processing / reprocessing, causing, together with relatively high temperatures, changes in the structure of properties of materials.

Lines 199-201: Knowing the reprocessability behaviour helps choosing the most suitable thermoplastic matrix / natural filler (fiber) partners among various candidates, to reach the requirements of the designed new materials and of the downcycled intended products.

Q4: PAG 8 line 324 - I ask the author to discuss better the degradation mechanisms that are involved during weathering and agening. There are some articles that try to explain how these happen. For example

Fortini A., Mazzanti V. Combined effect of water uptake and temperature on wood polymer composites, Journal of applied polymer science 135(35),46674, 2018

Badji, C., Beigbeder, J., Garay, H.Email Author, Bergeret, A., Bénézet, J.-C., Desauziers, V. Natural weathering of hemp fibers reinforced polypropylene biocomposites: Relationships between visual and surface aspects, mechanical properties and microstructure based on statistical approach, Composites Science and Technology Volume 167, 20 October 2018, Pages 440-447

A4: Discussion was added, in lines 472-478 and 487-490, as following:

Lines 472-478: Swelling / shrinking of anisotrope natural fillers occurs inside thermoplastic matrices at water absorption / desorption cycles. Internal tensions appear due to the difference between reversible dimensional changes of fillers and plastic deformation of matrix, leading to stress-induced damages of the filler / matrix interface, and pull-out of fillers, with formation of voids [140]. Fillers occupy the available voids in subsequent swelling, increasing the interfacial tension. Eventually, after repeated absorption/desorption cycles under combined effect of moisture and temperature, gaps appear around filler particles, leading to cracks inside material that extend with exposure.

Lines 487-490: Degradation during weathering occurs firstly in the amorphous regions, leading to relative increase of the crystalline phase. New crystalline domains appear by rearrangement of shorter chains resulted from the degradation of macromolecules in amorphous phase. Advanced weathering eventually also affects the crystalline phase [145].

Q5: Furthermore, it could be convenient for the reader to have a table with the references in which it is possible to quantify the trend of the mechanical properties as a function of degradation

A5: Literature reports presented here covers a broad range of materials and degradation with mainly qualitative discussion, therefore quantitative trend cannot be drawn.